# Effect of Coronary Collateral Supply on Left Ventricular Global Longitudinal Strain after Recanalization of Chronic Total Occlusion

**DOI:** 10.3390/diagnostics14182007

**Published:** 2024-09-11

**Authors:** Haci Ali Kurklu, Nil Ozyuncu, İrem Muge Akbulut Koyuncu, Kerim Esenboga, Turkan Seda Tan

**Affiliations:** 1Department of Cardiovascular Medicine, Ankara Etlik Research Hospital, 06170 Ankara, Turkey; hacialikurklu@gmail.com; 2Department of Cardiovascular Medicine, Ankara University School of Medicine, 06170 Ankara, Turkey; imakbulut@ankara.edu.tr (İ.M.A.K.); kerimesenboga@yahoo.com (K.E.); skurklu@ankara.edu.tr (T.S.T.)

**Keywords:** chronic total occlusion, global longitudinal strain, coronary collateral supply, coronary artery disease

## Abstract

Percutaneous coronary intervention (PCI) to chronic total occlusion (CTO) is still a subject of debate. The primary goal of revascularization is to provide symptomatic relief and enhance left ventricular (LV) functions. Global longitudinal strain (GLS) is proven to be more sensitive than the ejection fraction (EF), especially for subtle ischemic changes. The purpose of this study was to investigate the improvement in LV GLS after revascularization of symptomatic stable coronary patients with single-vessel CTO, categorized according to their collateral supply grades. Sixty-nine patients with successful CTO-PCI were grouped, according to their collateral supply grades, as well-developed (WD) and poor collateral groups and followed-up for 3 months. Basal characteristics were similar for both groups, except for a lower EF (*p* = 0.04) and impaired GLS (*p* < 0.0001) in the poor collateral group. At the end of 3 months follow-up, symptomatic relief was similar in both groups (*p* = 0.101). GLS improvement reached statistical significance only for the poor collateral, not for the WD group (*p* < 0.0001 and *p* = 0.054, respectively). The EF did not change significantly in both groups. Poorly collateralized CTO lesions may not only result in baseline LV dysfunction, but also appear to carry potential for recovery after revascularization. This may not be the case for WD collaterals.

## 1. Introduction

A chronic total occlusion (CTO) refers to a completely occluded coronary artery with TIMI 0 flow with a duration of ≥3 months [1]. In studies involving patients undergoing invasive coronary angiography, the prevalence of at least one CTO has been reported to be 15–20% [2]. Despite the high frequency of CTO, percutaneous recanalization has traditionally been considered as a technical challenge. Today, with recent technological advances, such as recanalization techniques and equipment, coupled with advancing operators’ expertise, the success rate of CTO recanalization has risen to 80–90% at specialized CTO referral centers [3].

The benefits of CTO revascularization via percutaneous coronary intervention (PCI) remain controversial. Today, the primary objective of CTO-PCI on top of medical treatment is to achieve symptom improvement. Uncertainties regarding the reduction in major adverse cardiovascular events (MACEs) persist due to the existing trials being underpowered and yielding conflicting results [4]. Moreover, several studies showed no functional improvement in left ventricle (LV) function after revascularization [5,6]. These studies, however, assessed the systolic function by using the LV ejection fraction (LVEF) and LV diameters by conventional echocardiography. In speckle-tracking echocardiography (STE), image processing algorithms are applied to routine two-dimensional echocardiographic images to identify small, stable myocardial speckles within a defined region of interest. By tracking these speckles frame-by-frame over the cardiac cycle, the distances between them or their spatiotemporal displacement (regional strain velocity vectors) provide valuable information about global and segmental myocardial deformation, so-called myocardial strain analysis [7]. The LV global longitudinal strain (GLS) is the novel parameter to evaluate LV systolic function with superior reproducibility compared to LVEF and has been validated for its efficacy in detecting myocardial ischemia [8]. Myocardial dysfunction can manifest even when the EF is preserved, potentially correlating with impaired LV longitudinal deformation [9]. In patients with preserved LVEF, the relationship between GLS and EF was shown to be curvilinear, in contrast to the linear relationship in reduced LVEF [10]. Consequently, GLS has a superior ability to discriminate and detect subclinical myocardial dysfunction when LVEF is within normal ranges [11]. The ischemic cascade begins with impairment in the watershed subendocardial layer, where longitudinally oriented fibers are predominant, so it is not surprising that GLS abnormalities have been reported even in patients with subclinical ischemia [7]. GLS was shown to be both an independent predictor of significant coronary artery disease and a significant predictor of outcomes during long-term follow-up in patients with chronic ischemic cardiomyopathy [11,12,13]. Furthermore, several studies have shown that the LV GLS improves after the recanalization of the CTO [14,15].

In the presence of a CTO, the development of collateral vessels can mitigate myocardial necrosis and sustain contractile function within the region distal to the occlusion. The presence of a well-developed (WD) coronary collateral circulation (Rentrop 2–3 classification) has been demonstrated to preserve LV systolic functions in CTO patients [16,17]. Moreover, improved coronary collateral circulation can diminish infarct size, thereby lowering the likelihood of ventricular aneurysm formation, reducing future cardiovascular events, and enhancing survival rates [18]. Although a WD collateral circulation system may adequately perfuse the myocardium at rest, it could potentially be insufficient during periods of increased demand, leading to ischemia [19].

In the literature, LV functions after the recanalization of CTO have been investigated irrespective of coronary collateral circulation. We hypothesized that among patients with CTO and WD coronary collateralization, there may be no significant improvement in LV functions following revascularization. This could be attributed to the presence of sufficient blood supply at rest, potentially limiting the enhancing effects of recanalization on LV function. Thus, we employed LV GLS analysis in our study to assess whether CTO recanalization provided benefits in restoring LV function. We compared the outcomes between patients with WD and poor coronary collateralization.

## 2. Materials and Methods

### 2.1. Patient Data

In this prospective, multicenter observational study, a cohort of 90 symptomatic patients who planned to undergo CTO revascularization of a single major coronary artery were consecutively screened between January 2021 and February 2023. A total of 69 patients were enrolled in our study, and the patient flow chart is presented in Figure 1. Symptomatic patients with typical angina pectoris according to the Canadian Cardiovascular Society (CCS) under optimum medical therapy (OMT) were included in the study [20]. Patients with atypical symptoms, such as atypical angina, exercise dyspnea, or intolerance, were also included if there was stress-induced ischemia in ≥10% of the global myocardial region in myocardial perfusion scintigraphy (MPS). Patients with recent revascularization for non-CTO lesions, including acute coronary syndromes, were eligible if 6 weeks had passed after successful revascularization. Patients with acute coronary syndrome, unstable angina, ≥50% stenosis in a non-CTO artery, unsuccessful CTO-PCI, arrhythmia, moderate-to-severe valvular disease, cardiomyopathies, akinesia or aneurysm of the target artery area, and patients with reduced EF (EF ≤ 40%) were excluded from the study.

The CTO of the major epicardial artery was defined as complete occlusion with TIMI 0 flow for at least 3 months’ duration in a major coronary artery with a vessel diameter of ≥2.5 mm. The operator judged the suitability and the method of the planned revascularization of the CTO lesion after diagnostic coronary angiography. The procedural approach for CTO included either an antegrade or retrograde approach adhering to contemporary technical guidelines [3]. In cases where the initial PCI had failed, further treatment options, including medical or surgical therapy, were pursued based on clinical requirements. Medical histories, encompassing all clinical and demographic data, including cardiovascular risk factors such as hypertension (HT), diabetes mellitus (DM), dyslipidemia, and smoking, were extracted from electronic medical records and patient interviews. Laboratory results obtained within 24 h prior to coronary angiography were recorded. Patients were followed-up at the first and third months after CTO recanalization. During this follow-up visit, they underwent clinical and echocardiographic evaluations, and were also interviewed about their anginal symptoms. Angina pectoris was graded in accordance with the CCS system and grading of the score was performed at the initial visit, and was also questioned and recorded during the consecutive visits by the attending physician [20]. All subjects gave their informed consent for inclusion before they participated in the study. The study was conducted in accordance with the Declaration of Helsinki and the study protocol was reviewed and approved by the local ethics committee.

### 2.2. Transthoracic Echocardiography

Screened patients who met the clinical criteria for study inclusion underwent two-dimensional echocardiographic imaging 24 h prior to the CTO-PCI. Sequential echocardiographic measurements were conducted in the first and third months following the recanalization. Two experienced cardiologists (TST, NO) performed the transthoracic echocardiography (TTE) using the Vivid E9 imaging system (GE Medical Systems, Chicago, IL, USA), and measurements were obtained following the recommendations of the American Society of Echocardiography [8]. LV dimensions were measured in the parasternal long-axis view at end-systole and end-diastole. LV EF was calculated using the modified Simpson method from four-chamber views.

### 2.3. Left Ventricular Global Longitudinal Strain

An automated function imaging (AFI) method was used in the Echo Pack imaging workstation (GE, Echo Pack imaging systems) to obtain 4-chamber, 2-chamber, and APLAX images with a frame rate of 50–70 fps. Two-dimensional STE measurements of the LV strain were obtained according to the recommendations of the 2015 EACVI/ASE consensus document [8]. LV strain measurements were omitted from the analysis if the left ventricular image quality was deemed inadequate or if adjusting the region of interest (ROI) did not enhance tracking quality. An LV GLS of >−20% was defined as impaired LV GLS by the guidelines. Moreover, the consensus is that the variations among vendors and software packages are still too large to recommend universal normal values [8] (Negative values in GLS are consigned to longitudinal shortening in systole. Greater degrees of deformation translate to numerically lower strain values and practically higher absolute values of GLS.) The lower limit of normality for GLS was considered as −16% based on population analysis [11,21].

### 2.4. Coronary Angiography

An interventional cardiologist performed coronary angiography according to standard procedures. Following diagnostic coronary angiography, all patients underwent heart-team discussion. Patients with typical anginal symptoms under OMT were enrolled directly if they had no exclusion criteria. For other symptomatic patients, MPS was ordered to detect ischemia in the target area, as these patients presented with atypical angina or exercise intolerance and required further diagnostic clarification. Patients who were indicated for revascularization by the heart team underwent PCI, performed by experienced interventional cardiologists using femoral access. In accordance with their expertise, the medical team recanalized the CTO, using specialized wires, balloons, and microcatheters. All patients were revascularized with drug-eluting stents. Preoperative, perioperative, and follow-up medical management, including the antiplatelet regimen, adhered to the current guidelines [22]. Angiographic success was defined as achieving a final angiographic residual stenosis of less than 20% by visual estimate and TIMI III flow following the implantation of stents [23]. Two blinded interventional cardiologists classified coronary collaterals according to the Cohen–Rentrop classification, which was graded as follows: grade 0 = no visible collaterals, grade 1 = the filling of the side branch via collateral vessels without visualization of the epicardial segment, grade 2 = the partial filling of the epicardial coronary artery, and grade 3 = the complete filling of the epicardial coronary artery [24]. Enrolled patients were classified into WD (grades 2 and 3) and poor (grades 0 and 1) coronary collateral groups [25]. Figure 2 shows representative images of a randomly selected study patient who had WD collaterals and underwent successful CTO recanalization, followed-up with consecutive LV GLS measurements recorded at baseline and at the third month.

### 2.5. Statistical Analysis

Baseline characteristics were presented as the mean ± SD for continuous variables or percentages for categorical variables. Changes in LV end-diastolic diameter (LVEDD), LV end-systolic diameter (LVESD), LVEF, and LV GLS data from baseline to the first and third months post-procedure were evaluated using one-way repeated-measures analysis of variance (ANOVA). Additionally, a mixed-design ANOVA test was used to compare the repeated measures in two collateral groups (between subjects and within subjects). The linear regression method was used to evaluate the parameters associated with changes in GLS. In the multivariate analysis, a hierarchical model was utilized to identify independent predictors of GLS improvement. A value of *p* < 0.05 was considered statistically significant. All data were analyzed using IBM SPSS Statistics version 26 (SPSS Inc., Chicago, IL, USA).

### 2.6. Interobserver and Intraobserver Variability

Images from 20 randomly selected patients were independently measured by a second blinded observer to evaluate interobserver variability. Subsequently, the first observer re-measured the same 20 patients’ images at least four weeks after the initial measurements were taken. The same vendor was utilized throughout the trial, and both observers had been working in the imaging department for eight years. Interobserver and intraobserver variability were analyzed using the intraclass correlation coefficient (ICC).

## 3. Results

### 3.1. Baseline Characteristics

The study enrolled 69 patients (mean age 64.9 ± 9.4, 81% male) who underwent CTO-PCI of a single epicardial coronary artery. According to the coronary collateral supply to the CTO, patients were classified as having WD (36 patients, 52%) or poor (33 patients, 48%) collaterals. All patients enrolled in the study had typical angina pectoris or significant ischemia, as demonstrated in the case of patients with angina-equivalent symptoms. According to the CCS angina grading system, 28 (41%) patients experienced grade 2 angina, 34 (49%) patients had grade 3 angina, and 7 (10%) patients reported grade 4 angina pectoris. Grade 1 angina pectoris was not observed in the study population. Patients in the poor coronary collateral group demonstrated significantly more severe angina levels when compared to the WD collateral group (*p* = 0.02). Demographic and clinical characteristics, laboratory results, and cardiovascular risk factors for the study cohort, together with the two groups according to the collateral supply, are demonstrated in Table 1. All patients were on appropriate doses of statins, beta-blockers, and angiotensin receptor blockers (ARBs) or angiotensin-converting enzyme inhibitors (ACEIs), as well as on dual antiplatelet therapy, in accordance with the clinical guidelines throughout the study [22].

### 3.2. Coronary Angiography

All patients were symptomatic and had a CTO at one of their major coronary arteries. None of the patients exhibited secondary arterial stenosis of ≥50%, and all underwent intervention for their single-vessel CTO lesion. Table 1 summarizes the angiographic characteristics of the patients, encompassing details such as coronary arteries with CTO, history of myocardial infarction (MI), prior PCI, and coronary artery bypass surgery (CABG). Both the WD and poorly developed collateral groups were statistically similar according to their angiographic characteristics. All revascularization procedures were considered successful by the attending interventionists according to the criteria defined in the Methods Section. No complications, such as pericardial tamponade, coronary rupture, MI, or death, were observed perioperatively. After revascularization, symptomatic relief of angina or equivalents was reported by 54 patients at the end of the third-month visits (75% of patients in the WD group, 81% of patients in the poor collateral group, *p* = 0.101). Fifteen patients reported no improvement in their symptoms, with nine patients in the WD collateral group and six in the poor collateral group. None of the patients experienced an exacerbation of their symptoms.

### 3.3. Echocardiographic Measurements

All patients underwent baseline TTE within 24 h prior to the PCI. The mean basal LVEDD and LVESD were 51.03 ± 6.04 mm and 29.02 ± 4.19 mm, respectively, and there were no statistically significant differences between the collateral groups. None of the patients had EF < 40%, and the mean basal EF of the study group was 53.2 ± 3.86. Although the difference was slight, the basal EF of the WD collateral group was statistically higher than that of the poor collateral group (54.2 ± 3.4 vs. 52.2 ± 4.1, respectively, *p* = 0.04). The baseline LV GLS values of the study population were impaired, with a mean value of −13.8 ± 1.66. The baseline LV GLS of the WD collateral group was significantly lower (meaning less impaired, as the absolute value is higher) than that of the poor collateral group (−14.8 ± 1.16 vs. −12.8 ± 1.47, respectively, *p* < 0.0001) (Table 1).

Echocardiographic parameters were re-monitored at the first and third months following PCI. One-way ANOVA was used to analyze repeated measures of LVEDD, EF, and LV GLS. Significant improvement was observed in GLS across baseline, first, and third-month measurements (−13.8 ± 1.66 vs.−15.3 ± 1.29 vs. −15.5 ± 1.26, respectively, *p* = 0.0001). However, there was no improvement in the EF (53.2 ± 3.86 vs. 53.3 ± 3.8 vs. 53.4 ± 3.7, respectively, *p* = 0.7) and in the LVEDD values (51.03 ± 6.04 mm vs. 50.9 ± 5.9 mm vs. 50.8 ± 5.9 mm, respectively, *p* = 0.06) (Table 2).

A mixed ANOVA method was employed to assess differences between the WD and poor collateral groups across repeated measures, as well as within the repeated measures of each group. There was no significant difference in the EF between the WD and poor collateral groups (*p* = 0.56). Furthermore, no significant difference was observed within the repeated measures of the EF in the WD and poor collateral groups individually (Table 3). LV GLS values differed significantly between the WD and poor collateral groups at baseline, first-, and third-month follow-ups, with the poor collateral group revealing significant improvement when compared to the WD group (WD collateral GLS: −14.8 ± 1.16; −15.2 ± 1.13; −15.2 ± 1.1, respectively, vs. poor collateral GLS: −12.8 ± 1.47; −15.5 ± 1.45; −15.9 ± 1.33, respectively, *p* < 0.0001). LV GLS was found to improve significantly from baseline to the first month, and significance was preserved at the third month in the poor collateral group. The partial effect size of GLS was calculated to be 0.72 and was interpreted as a medium effect size. There were no significant differences between the baseline, first-month, and third-month measures in the WD collateral group. Table 3 and Figure 3 represent the mixed ANOVA measurements in detail.

A multivariate analysis was conducted to identify the independent predictors of GLS improvement. Age, gender, baseline EF, prior MI, prior PCI, and the level of coronary collateral supply (WD or poor) were variables identified by the linear regression method and were analyzed. Only the poor collateral circulation was found to be strongly associated with the improvement in GLS from baseline to the third month (B: −2.773 [95% CI: −3.261 to −2.286], *p* < 0.001).

The intraobserver (ICC: 0.97, 95% CI: 0.93–0.98) and interobserver (ICC: 0.94, 95% CI: 0.88–0.96) agreements of the LV strain measurements were excellent.

## 4. Discussion

In this trial of patients with symptomatic stable coronary artery disease and single-vessel CTO, successful PCI procedures resulted in progressively improved LV function measured by LV GLS in the poor collateral supply group. However, no such improvement was observed in the WD collateral supply group. There was no significant improvement in the EF in both collateral groups. The study cohort consisted of a highly selective group of preserved EF patients with a single-vessel CTO and no significant lesions in non-CTO vessels. Additionally, these patients exhibited either typical anginal symptoms or significant ischemia in the CTO territory, without akinesia or dyskinesia indicative of non-viable myocardium. Consequently, our study demonstrated the impact of CTO revascularization on LV strain values based on collateral supply levels. To the best of our knowledge, no prior articles have explored this specific topic.

The decision to recanalize a CTO presents a significant dilemma for cardiologists. A recent systematic review and meta-analysis compared the use of OMT and PCI in patients with known CTOs and revealed a potential improvement with PCI in overall mortality, cardiac death, repeat revascularization, and MACEs; however, these improvements did not reach statistical significance [26]. More randomized controlled trials (RCTs) are needed for a definitive answer. Despite this, there is currently no evidence to advise against PCI in CTO, and there may potentially be some benefits. Today, angina or angina-equivalent symptoms resistant to OMT remain the main indication for CTO recanalization, as MACE reduction at follow-up remains uncertain based on the currently available data [4]. Evidence supporting CTO-PCI as an effective tool for symptom relief and improvement in quality of life is primarily based on three randomized controlled trials [27,28,29]. In the design of our trial, we enrolled the symptomatic patients on OMT and verified significant ischemia for the ones with atypical or possibly angina-equivalent symptoms, consistent with the recent guidelines [22]. Symptomatic improvement after revascularization was achieved in 78% of the study population, with statistically similar outcomes observed between the subgroups of collateral supply. The high anginal recovery rate following single-vessel CTO revascularization may underscore the importance of appropriate patient selection.

The possibility of functional recovery of the LV is another consideration for CTO revascularization; however, its role remains subject to debate [4,30]. In a meta-analysis of 34 studies including symptomatic patients with a large ischemic burden or LV dysfunction, it was found that CTO-PCI induced an improvement of 4.44% in the EF (*p* < 0.01) [31]. CTO-PCI may show benefits, especially for patients with more severe LV dysfunction and significant myocardial perfusion defects in viability, as shown by previous trials [32,33,34]. However, these findings were not confirmed in the randomized REVASC trial in which baseline LVEF was only mildly to moderately reduced in both the OMT and CTO-PCI groups. The functional recovery in the CTO territory revealed no differences between the invasive and non-invasive treatment groups by magnetic resonance imaging (MRI) (change in EF: 0.9 vs. 0.7, *p* = 0.79) [6]. Schumacher et al., consistent with the findings of the ISCHEMIA trial, suggested that patients with stable coronary disease and impaired contractility constituted a subgroup where extensive ischemia relief via coronary revascularization may provide a protective effect, which is less pronounced in patients with preserved LV function [35,36,37]. In the TOSCA study, basal LV dysfunction was shown to be an independent predictor of improvement in the EF after CTO-PCI [38]. In our trial, the study population had preserved their EF and were symptomatic under OMT. We evaluated myocardial viability exclusively in patients with atypical or subtle symptoms, as those presenting with typical anginal symptoms were already presumed to have significant myocardial jeopardy. Myocardial revascularization guidelines do not suggest a routine analysis of viability before PCI [22]. On the other hand, patients with CTO and inadequate collateral circulation may benefit from assessment of myocardial viability to determine the most suitable candidates for PCI [39]. In our selective cohort of patients with the ischemic burden of a single-vessel CTO lesion, no significant changes in the EF were observed at the 3-month follow-up after revascularization, either in the entire group or between the groups, based on their CTO collateral supply. However, when evaluating our results, it should be considered that the 3-month follow-up period might have been insufficient for demonstrating improvement in the EF.

Conventional echocardiography allows the identification of significant LV dysfunction, but not subclinical dysfunction and recovery. The two-dimensional (2D) speckle-tracking of the LV strain is an established technique used in the evaluation of detailed LV systolic functions. GLS represents the overall change in myocardial length during a cardiac cycle. It has been shown to sensitively reflect the damage to subendocardial myocardium, the region most prone to ischemic changes [40]. Regarding the LV contractile functions, LV GLS can provide more accurate and earlier information than the classical EF measurements, with demonstrated feasibility and reproducibility [8]. Subtle changes in contractility, whether decreases due to damage or increases due to recovery, can be effectively detected through strain analysis [41,42]. In the literature, successful CTO-PCI was shown to improve LV function measured by strain echocardiography, especially if viability and ischemia were evident [43,44]. Wang et al. demonstrated that LV EF values tended to improve after the third month of CTO-PCI; however, GLS values began to recover as early as the first day after successful revascularization [15]. Sotomi et al. demonstrated a significant improvement in GLS, but not in the EF, among patients undergoing single-vessel CTO-PCI, with significance beginning from the first day post-procedure and persisting at three months (GLS pre-PCI, 1-day post-PCI, and 3-months post-PCI were given, respectively: −12.8 ± 4.2, −14.3 ± 4.1, −14.3 ± 4.4, *p* = 0.023) [14]. GLS was shown to be a reliable tool to reveal myocardial recovery after revascularization, especially in preserved EF patients when compared to EF values [11]. An additional advantage was the demonstration of myocardial recovery within a relatively short period of time. Our study investigated the potential impact of collateral supply grade on GLS in patients undergoing single-vessel CTO-PCI in a 3-month follow-up period. We found that GLS values improved in the poor collateral group. In the literature, the clinical impact of GLS improvement following revascularization on cardiovascular outcomes remains a subject of debate. In a small single-center trial, non-improvement in GLS at the third month did not alter the cardiovascular outcomes in post-MI patients [45]. Further larger studies are needed on the impact of GLS changes on cardiovascular outcomes.

In our trial, GLS values improved only in the poor collateral group, not in the WD collateral group. Poor collateral circulation was the only significant determinant of GLS improvement in multivariate analysis. It was demonstrated that after CTO-PCI, the regional longitudinal strain had improved significantly not only in the CTO area but also in the CTO collateral donor vessel area [14]. This led to the hypothesis that both the collateral recipient CTO area and the collateral donor vessel area could be at ischemic risk. This may be particularly true for diseased donor vessels, where collateral supply to the CTO area may lead to a reduction in the blood flow of the donor vessel itself, known as the “steal phenomenon” [46]. Our study patients had only one CTO lesion, and the remaining coronaries were free from significant stenosis (≥50% stenosis in non-CTO arteries were excluded). Therefore, it was anticipated that collateral donor vessel flow would not fall below ischemic thresholds, and regional strain in the donor vessel territory would remain unchanged after revascularization. This could explain the lack of improvement in GLS in the WD collateral group, as both the collateral donor and recipient coronary territory were expected to receive adequate blood flow to maintain contractile function. However, inefficient coronary supply under increased demand appeared evident, given that all patients were either symptomatic or had significant provoked ischemia. It is suggested that long-term follow-up trials with larger patient cohorts are necessary to further clarify the differences in regional myocardial improvement based on the degree of collateral supply.

Patients with CTO typically develop collateralization of the distal vessel at varying degrees. This can contribute to the relief of ischemia and anginal symptoms, as well as to the preservation of ventricular function [16,17]. In a meta-analysis of 12 studies and 6529 patients, WD collateralization in coronary artery disease was associated with a higher survival rate and yielded a 36% reduction in mortality risk compared to poor collateralization [47]. Numerous studies have demonstrated that WD collateral circulation improved survival and LV function by maintaining adequate blood supply, thereby preserving metabolic function and preventing necrosis [48,49]. A strong correlation was observed between the degree of coronary collateral circulation and significantly smaller myocardial infarctions, along with a preserved LV EF [18,50]. Many physicians may view robust coronary collateral circulation as indicative of a favorable prognosis in patients with CTO, and, consequently, they may often recommend medical therapy over revascularization. Nevertheless, during exercise, the functional reserve of these collaterals is known to be limited, leading to ischemia in more than 90% of patients with well-collateralized occlusions [48]. In the context of CTO revascularization, there does not seem to be a corresponding improvement in prognosis according to the degree of collateral supply [51]. A plausible explanation for these results may stem from the grading technique typically employed for collateral assessment. The classical angiographic grading system, described by Rentrop et al., evaluates the effectiveness of collaterals in filling the occluded arterial segment rather than assessing the function and quality directly [24]. Although these anatomic methods are practical, they do not necessarily correlate with functional perfusion [46]. A study by Werner et al. revealed that there was no relationship between Rentrop collateral grade classification and collateral function assessment in CTO patients [52]. However, these functional assessments, determining flow and pressure indices, are technically challenging and operator-dependent. Today, the physiological assessment of collateral supply is not used in clinical practice and remains a research tool. In our study, Rentrop collateral grading was used and patients were grouped according to the supply grade. Our clinical and LV functional outcomes across groups at pre-PCI and post-PCI follow-ups appeared to align with the collateral grading scores. Patients with WD collateral supply had significantly higher EF and GLS values when compared to the poor supply group at the initial visit (EF: 54.2 ± 3.4 vs. 52.2 ± 4.1, *p* = 0.04, GLS: −14.8 ± 1.6 vs. −12.8 ± 1.47, *p* = 0.0001). This difference observed during the baseline examination may be attributed to the potentially larger area of hibernating myocardium in the poor collateral group. This mechanism can also explain the significant improvement in GLS observed at the post-PCI follow-ups, which was evident, again, only in the poor collateral group.

### Limitations

Our study has several limitations that must be considered. The patient volume was limited and included only those with successful PCI, which might overestimate the benefits of the procedure. Our follow-up period was relatively short and might have been insufficient to detect the effects of long-term PCI complications and improvements in echocardiographic parameters. While the follow-up time was ideal for detecting functional changes via GLS, EF analysis might have shown significant changes with a longer follow-up period. Additionally, we did not employ a validated patient-based angina questionnaire to assess anginal symptoms, which may have led to an overestimation or underestimation of symptom severity. However, a physician-based anginal scoring system, such as that provided by the CCS, could provide results that are non-inferior, and possibly superior, as this method has been widely utilized in previous studies. It has been demonstrated that physician interpretation of an individual’s symptoms plays a crucial role in evaluating the severity of their coronary artery disease when compared to patient-based questionnaires [53].

## 5. Conclusions

The recanalization of CTO lesions remains a controversial topic. In a carefully selected group of CTO patients, successful revascularization may lead to an improvement in LV function detected by strain analysis, in addition to a high incidence of symptomatic relief. We concluded that the revascularization of CTO lesions with WD collaterals may result in no LV functional recovery if the collateral donor vessel is free of any significant lesion that is capable of causing a steal phenomenon. However, significant symptomatic relief was revealed for both collateral groups. Poorly collateralized CTO lesion territories not only resulted in baseline LV dysfunction, but also appeared to carry the potential for recovery after revascularization. We believe our results highlight the importance of careful patient selection for CTO revascularization and the value of GLS analysis for the detection of subtle LV functional changes. Our findings may encourage further research on CTO revascularization with a focus on collateral supply grading and the use of strain analysis for functional assessment in studies with extended follow-up periods and larger patient populations.

## Figures and Tables

**Figure 1 diagnostics-14-02007-f001:**
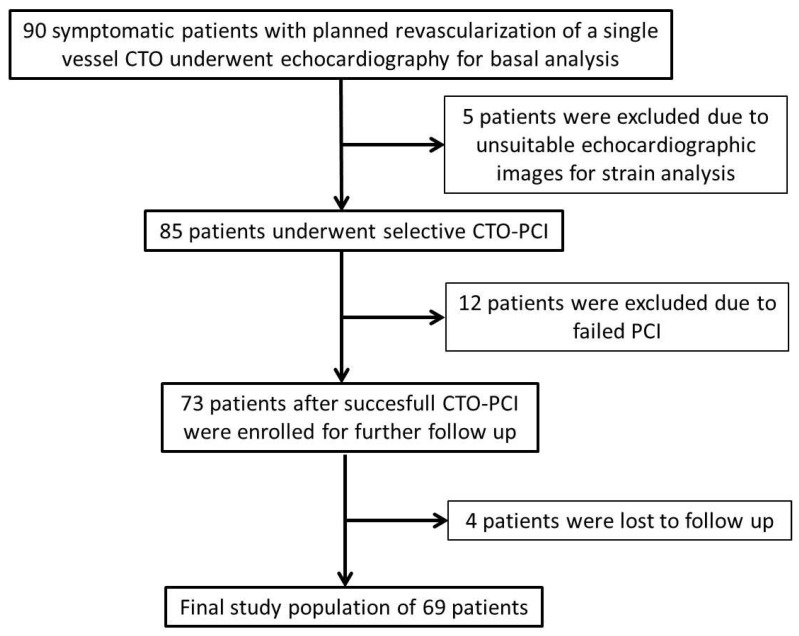
Patient flow chart. CTO: chronic total occlusion, PCI: percutaneous coronary intervention.

**Figure 2 diagnostics-14-02007-f002:**
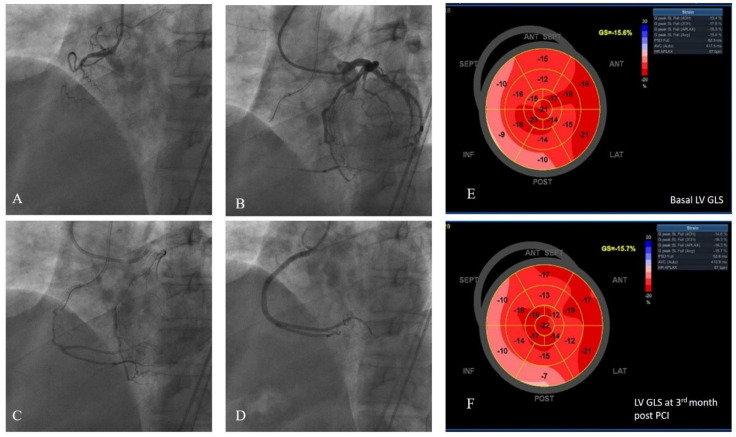
LV GLS strain and coronary angiography views of a random study patient. (**A**) Chronic total occlusion of RCA. (**B**,**C**) Dual injection demonstrated the presence of grade 3 collateral circulation from the left coronary arteries to the RCA. (**D**) The RCA was successfully wired, two drug-eluting stents were implanted, and TIMI 3 flow was demonstrated. (**E**) Pre-PCI average LV GLS was measured to be −15.6%. (**F**) Post-PCI third month LV GLS was measured to be −15.7%. LV GLS: left ventricular global longitudinal strain, RCA: right coronary artery, PCI: percutaneous coronary intervention.

**Figure 3 diagnostics-14-02007-f003:**
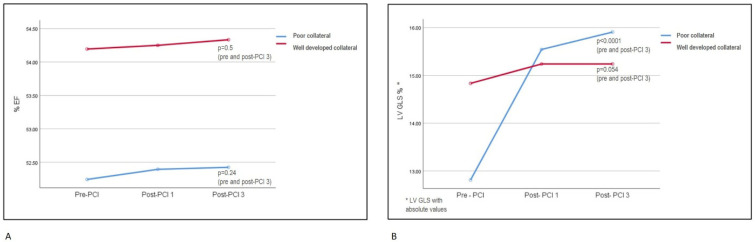
Mixed-design ANOVA analysis of repeated measures of EF and LV GLS in WD and poor collateral groups. (Please note that absolute values of LV GLS measurement were used in the chart and *p*-values in the figure were provided for the pre-PCI and third month post-PCI periods; detailed *p*-values were presented in Table 3.) (**A**): Repeated measures of EF did not show any differences between collateral groups and also within groups. (**B**): LV GLS improvement in poor collateral group was significantly higher than in WD group. All LV GLS values of consecutive follow-ups showed improvement in the poor collateral group, but not in the WD group, when examined separately. EF: ejection fraction; LV GLS: left ventricular global longitudinal strain; PCI: percutaneous coronary intervention; Post-PCI1: first month after PCI; Post-PCI3: third month after PCI.

**Table 1 diagnostics-14-02007-t001:** Baseline characteristics of the study population and comparison of subgroups according to coronary collateral circulation.

Characteristics	Study Population (N = 69)	Well-Developed Collateral (N = 36)	Poor Collateral (N = 33)	*p*-Value
**Age**	**64.9 ± 9.4**	64.7 ± 9.7	65.1 ± 9.3	0.9
**Male n, (%)**	**56 (81.2)**	28 (78)	28 (85)	0.5
**BSA (m^2^)**	**1.87 ± 0.15**	1.8 7 ± 0.1	1.86 ± 0.2	0.9
**HT n, (%)**	**50 (72.5)**	27 (75)	23 (70)	0.8
**DM n** **, (%)**	**41 (59.4)**	22(61)	19 (58)	0.8
**HL n, (%)**	**28 (40.6)**	12 (33)	16 (49)	0.2
**Smoking n, %**	**37 (53.6)**	18 (50)	19 (58)	0.63
**Family history n,%**	**22 (31.9)**	9(25)	13 (40)	0.3
**CCS angina grading (I/II/III/IV)**	**0/28/34/7**	0/24/18/1	0/14/16/6	0.02
**Laboratory results**				
**Fasting glucose mg/dL**	**101.1 ± 10.4**	104 ± 8.9	98.1 ± 11.2	0.05
**Hemoglobin g/dL**	**14.0 ± 2.13**	14.05 ± 2.1	14.1 ± 2.2	0.93
**Platelet**	**249 ± 75.05**	259.4 ± 85.8	246.7 ± 57.8	0.5
**Creatinine (mg/dL)**	**0** **.89 ± 0.** **19**	0.92 ± 0.18	0.86 ± 0.21	0.3
**TC mg/dL**	**192 ± 51.4**	191.3 ± 51.4	193.5 ± 48.8	0.9
**LDL-C mg/dL**	**125 ± 43.2**	120.5 ± 40.2	129.9 ± 46.4	0.4
**HDL-C (mg/dL)**	**43.3 ± 13.0**	43.6 ± 14.8	43.1 ± 11.01	0.9
**Echocardiography**				
**LVEDD (mm)**	**51.03 ± 6.04**	50.8 ± 5.6	50.9 ± 6.4	0.94
**LVESD (mm)**	**29.02 ± 4.19**	28.9 ± 4.2	29.2 ± 4.2	0.73
**EF** **%**	**53.2 ± 3.86**	54.2 ± 3.4	52.2 ± 4.1	0.04
**LV GLS %**	**−13.8 ± 1.66**	−14.8 ± 1.16	−12.8 ± 1.47	0.0001
**Coronary angiography**				
**CTO artery (LAD/CX/RCA)**	**27/11/31**	13/6/17	14/5/14	0.9
**Prior CABG n, (%)**	**0**	0	0	-
**Prior PCI to** **target coronary artery n, (%)**	**33 (47.8)**	18 (50)	15 (46)	0.8
**Prior PCI (all) n, (%)**	**37 (53.6)**	18 (49)	19 (51)	0.4
**Prior MI in CTO area**	**13 (18.8)**	9 (25)	4 (12)	0.2
**Prior MI (all) n, (%)**	**17 (24.6)**	9 (25)	8 (24)	1

Data are expressed as means ± SD, as n (%). BSA: body surface area; HT: hypertension; DM: diabetes mellitus; HL: hyperlipidemia; CCS angina grading: Canadian Cardiovascular Society Grading of Angina Pectoris; TC: total cholesterol; HDL-C: high-density cholesterol level; LDL-C: low-density cholesterol level; LVEDD: left ventricular end-diastolic diameter; LVESD: left ventricular end-systolic diameter; EF: ejection fraction; LV GLS: left ventricular global longitudinal strain; CTO chronic total occlusion; MI myocardial infarction; CABG: coronary artery bypass grafting; PCI: percutaneous coronary intervention; LAD: left anterior descending artery; LCX: left circumflex artery; RCA: right coronary artery.

**Table 2 diagnostics-14-02007-t002:** Repeated measures of echocardiographic parameters (n = 69).

Echocardiographic Parameters	Pre-PCI	Post-PCI M1	Post-PCI M3	*p*-Value (All)	*p*-Value Pre-PCI vs. Post-PCI M1	*p*-Value Pre-PCI vs. Post-PCI M3	*p*-Value Post-PCI M1 vs. Post-PCI M3
LVEDD mm	51.03 ± 6.04	50.9 ± 5.9	50.8 ± 5.9	0.06	0.2	0.2	0.3
EF%	53.2 ± 3.86	53.3 ± 3.8	53.4 ± 3.7	0.7	0.1	0.3	0.08
LV GLS%	−13.8 ± 1.66	−15.3 ± 1.29	−15.5 ± 1.26	0.0001	0.0001	0.0001	0.001

PCI: percutaneous coronary intervention, Pre-PCI: parameters were obtained within 24 h prior to PCI, M1: first month, M3: third month. LVEDD: left ventricular end-diastolic diameter; EF: ejection fraction; LV GLS: left ventricular global longitudinal strain.

**Table 3 diagnostics-14-02007-t003:** Comparison of echocardiographic parameters according to coronary collateral classification (well-developed collateral, n = 36, poor collateral, n = 33).

Parameters Divided by Collateral Classification		Pre-PCI	Post-PCI M1	Post-PCI M3	*p*-Value (All)	*p*-Value Pre-PCI vs. Post-PCI M1	*p*-Value Pre-PCI vs. Post-PCI M3	*p*-Value Post-PCI M1 vs. Post-PCI M3
WD Collateral	EF%	54.1 ± 3.37	54.2 ± 3.3	54.3 ± 3.2	0.56	1.0	0.5	0.3
Poor Collateral		52.2 ± 4.14	52.4 ± 4.1	52.4 ± 4.09		0.09	0.24	1.0
WD Collateral	LV GLS%	−14.8 ± 1.16	−15.2 ± 1.13	−15.2 ± 1.1	0.0001	0.07	0.054	1.0
Poor Collateral		−12.8 ± 1.47	−15.5 ± 1.45	−15.9 ± 1.33		0.0001	0.0001	0.0001

PCI: percutaneous coronary intervention; Pre-PCI: parameters were obtained within 24 h prior to PCI; M1: first month; M3: third month; WD: well-developed; EF: ejection fraction; LV GLS: left ventricular global longitudinal strain.

## Data Availability

Our data is unavailable due to privacy or ethical restrictions.

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
