# Peer review of "Effect of Coronary Collateral Supply on Left Ventricular Global Longitudinal Strain after Recanalization of Chronic Total Occlusion"

_diagnostics, 2024, doi:10.3390/diagnostics14182007_

Round 1

Reviewer 1 Report

Comments and Suggestions for Authors

See pdf attached

Reviewer 2 Report

Comments and Suggestions for Authors

The authors evaluated LV GLS changes following PCI for single-vessel CTO in stable coronary patients. Patients were categorized by collateral supply (well-developed vs. poor) and followed for three months. Baseline characteristics were similar except for lower EF and impaired GLS in the poor collateral group. GLS significantly improved in the poor collateral group (p<0.0001) but not in the well-developed collateral group (p=0.054), with no significant change in EF. This suggests poor collateralization may allow LV functional recovery post-revascularization, unlike well-developed collaterals.

Congratulations to the authors on this interesting work addressing a highly relevant topic. However, there are several issues to consider:

Issues/Suggestions:

Lines 109-121: This information can be included in the supplementary materials, and it is necessary to add the correct reference to lines 80-81.

Were the patients enrolled consecutively? This should be specified in the methods, and a flow chart could be useful.

Line 137: Why was a cut-off of GLS -20 chosen? Strong literature suggests that a lower cut-off may be more appropriate for CAD patients, e.g., undergoing echo-stress or not (e.g., PMID: 37921718). Please discuss this in the discussion section as well.

Tables: Add a column for the entire cohort.

I suggest performing a multivariate analysis of predictors of strain improvement to assess potential confounding factors in addition to the presence of collateral coronary arteries (with or without WD).

A viability study was conducted before proceeding with CTO revascularization. This is a highly debated topic in the current scientific literature (cite 10.1111/echo.15854). I recommend expanding the discussion and limitations section on this matter. If a viability study was not performed, I suggest to tone down the discussion section accordingly. 

Comments on the Quality of English Language

It is recommended that abbreviations and typos be reviewed carefully.

Reviewer 3 Report

Comments and Suggestions for Authors

The authors present an insightful study on the impact of percutaneous coronary intervention (PCI) on left ventricular (LV) function in patients with chronic total occlusion (CTO). They highlight the sensitivity of global longitudinal strain (GLS) over ejection fraction (EF) in detecting subtle ischemic changes. The study reveals that significant GLS improvement occurs primarily in patients with poor collateral supply following successful PCI, suggesting that these patients may experience better LV function recovery compared to those with well-developed collaterals. While the findings offer valuable insights into the potential benefits of PCI in CTO patients, there are aspects of the study that could be further clarified to enhance its overall impact.

1.     It is essential to mention in the Introduction section that global longitudinal strain (GLS) has been identified as a significant predictor of outcomes during long-term follow-up in coronary artery disease (CAD) patients (cite PMID: 38284665 to support this important point).

2.     I suggest creating a flowchart of the study to better illustrate how many patients were initially assessed and how many were ultimately included in the final analysis. This would enhance the clarity of the study's methodology and improve the reader's understanding of the patient selection process.

3.     I suggest that the authors include the number of patients excluded and analyzed in the Results section. The Methods section is not the appropriate place for this information, and moving it to the Results would provide a clearer overview of the study's final sample size and any exclusions made during the analysis.

4.     Could the authors provide more detailed information regarding the medical therapy administered to the patients? It would be helpful to include additional data in Table 1.

5.     I suggest reducing the length of the Discussion section, particularly the initial literature review. This part currently makes the reading heavy and shifts the focus away from the main findings of the study.

6.     The authors should address, in more detail, the plausible reasons behind the results obtained in the Discussion section.

Comments on the Quality of English Language

Minor editing of English language required. Please standardize everything in American English.

Round 2

Reviewer 2 Report

Comments and Suggestions for Authors

The authors have addressed my previous comments, and I have no further questions.

Comments on the Quality of English Language

A final review for abbreviations and typos is needed.

Author Response

Comments 1: Thank you for your valuable contributions. A final review for abbreviations and typos is done. Thank you

Reviewer 3 Report

Comments and Suggestions for Authors

Thank you to the authors for the revisions made, which I believe have enhanced the quality of the final manuscript.

Author Response

Thank you for all your valuable contributions